

# Quercetin alleviates PM$_{2.5}$-induced chronic lung injury in mice by targeting ferroptosis

Shibin Ding, Jinjin Jiang and Yang Li

Public Health and Management, Jiangsu Vocational College of Medicine, Yancheng, Jiangsu, China

## ABSTRACT

**Background**. PM$_{2.5}$ is a well-known harmful air pollutant that can lead to acute exacerbation and aggravation of respiratory diseases. Although ferroptosis is involves in the pathological process of pulmonary disease, the potential mechanism of ferroptosis in PM$_{2.5}$-caused lung inflammation and fibrosis need to be further clarified. Quercetin is a phenolic compound that can inhibit ferroptosis in various diseases. Hence, this study explores the role of ferroptosis in lung injury induced by PM$_{2.5}$ in order to further elucidate the beneficial effect of quercetin and its underlying mechanism.

**Methods**. C57BL/6J mice were treated with either saline or PM$_{2.5}$ by intratracheal instillation 20 times (once every two days). Additionally, PM$_{2.5}$-treated mice were supplemented with two doses of quercetin. Lung injury, lipid peroxidation, iron content and ferroptosis marker protein expression and the Nrf2 signaling pathway were evaluated. *In vitro*, cell experiments were applied to verify the mechanisms underlying the links between Nrf2 signaling pathway activation and ferroptosis as well as between ferroptosis and inflammation.

**Results**. *In vivo*, PM$_{2.5}$ increased lung inflammation and caused lung fibrosis and increased lipid peroxidation contents, iron contents and ferroptosis markers in lung tissues; these effects were significantly reversed by quercetin. Additionally, quercetin upregulated the nuclear Nrf2 expression and downregulated Keap1 expression in lung tissues of PM$_{2.5}$-exposed mice. Quercetin decreased lipid peroxidation products, iron contents and ferroptosis levels and increased the nuclear translocation of Nrf2 and the degradation of Keap1 in PM$_{2.5}$-exposed BEAS-2B cells. Moreover, we found that quercetin and dimethyl fumarate markedly decreased lipid peroxidation production and ferroptosis by activating the Nrf2-Keap1 pathway in PM$_{2.5}$-exposed cells. Furthermore, quercetin reduced inflammatory cytokines and TGF-$\beta$1 in PM$_{2.5}$-exposed cells.

**Conclusion**. Our data suggested that Nrf2 is involved in ferroptosis in PM$_{2.5}$-induced lung injury, and quercetin can alleviate these adverse effects via activating Nrf2-Keap1 signaling pathway.

Corresponding author
Shibin Ding, 12137@jsmc.edu.cn

## INTRODUCTION

Exposure to ambient PM$_{2.5}$ is the most important environmental risk factor related to public health worldwide. It has been reported that air pollution caused more than one million premature deaths each year in China (*Burnett et al., 2018*; *Lelieveld et al., 2015*).

PM$_{2.5}$ is one of the most important air pollutants in most countries worldwide, especially in developing countries. The major sources of ambient PM$_{2.5}$ originate from vehicle emissions, mineral dust, and biomass incineration (*Manousakas et al., 2017*). PM$_{2.5}$ can be directly inhaled into the alveoli and enter the circulatory system, systemically spreading and impairing multiple systems (*Laing et al., 2010*; *Yang et al., 2020*). Considering that PM$_{2.5}$ has been confirmed the fifth important risk factor for mortality and is the greatest contributor to human health burdens worldwide (*Landrigan et al., 2018*), it is important to study effective strategies for reducing the health hazards associated with PM$_{2.5}$ exposure.

Epidemiological evidence has confirmed that ambient PM$_{2.5}$ exposure is highly correlated with the morbidity of patients with respiratory diseases (*Pope 3rd et al., 2019*). In addition, an increasing amount of evidence from rodent experiments showed that PM$_{2.5}$ exposure triggers an inflammation response in various types of respiratory cells and induces lung fibrosis (*He et al., 2017*; *Zhao et al., 2020*; *Zheng et al., 2018*). The World Health Organization (WHO) has stated that the daily limit of exposure to PM$_{2.5}$ should not exceed of 25 $\mu g/m^3$ concentration, for the safety of human health. However, PM$_{2.5}$ at low concentrations also causes certain public health risks (*Fann et al., 2012*). Thus, in current study, we want to study the effect of low concentrations of PM$_{2.5}$ on the lung, which is the target organ for air pollutants.

Iron-dependent ferroptosis involves lipid peroxidation and iron accumulation (*Dixon et al., 2012*). Ferroptosis potentially serve a novel target for diagnosis and therapy of many diseases, such as pulmonary diseases (*Tao, Li & Liu, 2020*), cancers (*Nie et al., 2021*) and cardiovascular diseases (*Wu et al., 2021*). Interestingly, in *in vivo* studies, inhibition of ferroptosis improved PM$_{2.5}$-caused acute and chronic lung injury (*Guo et al., 2022*; *Yan et al., 2022*). Thus, modulation of ferroptosis is considered a new method for reversing respiratory system damage induced by PM$_{2.5}$ exposure. Quercetin (Que) is a natural flavonoid with antioxidant effect, can exert suppression effect on ferroptosis in nonalcoholic fatty liver disease and type 2 diabetes (*Jiang et al., 2022*; *Li et al., 2020a*). Previous studies have reported that Que treatment could prevent the pathologic changes in mice with COPD phenotype (*Farazuddin et al., 2018*) and alleviate radiation-induced lung injury (*Verma et al., 2022*). Thus, we want to investigate whether Que could improve PM$_{2.5}$ exposure-caused lung injury through inhibiting ferroptosis.

In the current study, we aimed to explore the effect and potential mechanism of chronic exposure to low-concentrations of ambient PM$_{2.5}$ on lung injury. Moreover, we studied the beneficial functions and possible molecular mechanisms of Que treatment on PM$_{2.5}$ exposure-caused lung toxicology.

## MATERIALS & METHODS

### Materials

Que (99% purity) was obtained from Solarbio Biological Technology Co., Ltd. (Beijing, China). Antibodies purchased from Sanying Bio, Inc. (Wuhan, China) included rabbit anti-GPX4, rabbit anti-ACSL4, rabbit anti-Nrf2, rabbit anti-Keap1, rabbit anti-TGF-$\beta$1, rabbit anti-Collagen-I, rabbit anti-GAPDH, and rabbit anti-PCNA primary antibodies and

goat anti-rabbit antibody. FBS and DMEM were obtained from Corning (Corning, NY, USA).

## Animal care and PM$_{2.5}$ exposure procedure

The 6–8 weeks Male C57BL/6J mice (18–22 g) were purchased from a commercial company (Vital River Laboratory Animal Technology Co., Ltd., Beijing, China). Mice were housed in the experimental animal room (20−24 °C, 40%–60% humidity) and maintained under 12-hour light/dark cycle. Three mice were fed in each cage and all mice were allowed to drink and eat freely. Animal experimental protocols were approved by the Ethics Committee at the Jiangsu Vocational College of Medicine (ethics number: LLSQ-202104160001).

Twenty-four mice were randomly (random number table method) assigned into four groups (six mice in each group): (1) the control group; (2) the PM$_{2.5}$ group; (3) the PM$_{2.5}$-Que50 group; and (4) the PM$_{2.5}$-Que100 group. A previous study reported that 50 mg/kg·bw Que can protect against LPS caused lung injury in mice (*Chen et al., 2022*). Thus, 50 mg/kg·bw and 100 mg/kg·bw Que were used for animal experiments. The mice in the PM$_{2.5}$-Que groups exposed to PM$_{2.5}$ (5 mg/kg·bw, once every two days, total 20 times) by intratracheal instillation and simultaneous daily supplemented with Que (50 and 100 mg/kg·bw) by oral gavage for 60 days. The animals in the control group were given intratracheal instillation of saline and oral gavage with 0.1 mL of deionized water daily. The dosage of PM$_{2.5}$ used in this study was estimated based on the interim target-1 for the annual mean PM$_{2.5}$ concentration (35 $\mu$g/m$^3$), which was suggested by the WHO air quality guidelines (*Jiang et al., 2021a*). Moreover, mouse respiratory times and the uncertainty factor were considered to calculate the dose of PM$_{2.5}$ (*Jiang et al., 2021a*). After PM$_{2.5}$ exposure (20 times intratracheal instillation), the mice were sacrificed using a 1% Pentobarbital Sodium for intraperitoneal anesthesia (*Lv et al., 2020*) and cervical dislocation according to the American Veterinary Medical Association (AVMA) Guidelines on Euthanasia.

## Sampling section

Blood samples were removed *via* heart puncture, and lung tissues were dissected and weighed. The lung tissues of the mice were quickly isolated. The right lungs were used for lung bronchoalveolar lavage. A part of left lungs was placed in 4% paraformaldehyde solution for histological observation and the other lungs were at −80 °C for further study.

## PM$_{2.5}$ collection

Cumulative PM$_{2.5}$ was collected with an air sampler (TE-6070C; Tisch Environmental, Cleves, OH, USA). The filters were maintained at −80 °C. To obtain PM$_{2.5}$, the filters loaded with PM$_{2.5}$ were cut into many pieces (2 cm ×2 cm) and sonicated (*Jiang et al., 2020*). PM$_{2.5}$ samples were suspended in saline and sonicated before intratracheal instillation.

## Lung bronchoalveolar lavage of the mice

After the last PM$_{2.5}$ exposure, the right lungs were washed with 0.5 ml of PBS 5 times. Collected bronchoalveolar lavage fluids (BALF) were centrifuged at 875× g/min for 10 min to pellet cells (*Ding et al., 2019*). The supernatant from the BALF was collected for pro-inflammatory cytokine analysis.

## Cell culture

BEAS-2B cells were obtained from the KUNMING Cell Bank (Kunming, China). BEAS-2B cells were cultured in a DMEM with 10% FBS and 100 units/mL penicillin-streptomycin and maintained at 37 °C in an incubator with 5% $CO_2$. Cell viability was determined by CCK-8 (DOJINDO, Shanghai, China). BEAS-2B cells were exposed with $PM_{2.5}$ or Que for 24 h to establish the treatment dose. In the previous study, 10 µmol/L Que was used in the subsequent study (*Lee & Yoo, 2013*). Thus, BEAS-2B cells were treated with different doses of Que (0, 5, 10, 20 and 40 µmol/L) to assess the cell viability. BEAS-2B cells were simultaneously treated with Que, 10 µmol/L ferrostatin-1 or the Nrf2-specific agonist dimethyl fumarate (DMF, 20 µmol/L; APEx Bio, Houston, TX, USA) in the presence of $PM_{2.5}$ for 24 h.

## Determination of pro-inflammatory cytokines

The levels of interleukin-1$\beta$ (IL-1$\beta$), TNF-$\alpha$ and interleukin-6 (IL-6) were determined by commercial ELISA kits (MLBio, Inc., Shanghai, China). The detected methods as follows: all kits were kept at room temperature (20−25 °C) for 30 min before use. Then samples were measured according to protocol and optical density of each sample wall was measured at 415 nm using the standard microplate reader (Enspire; PerkinElmer, Waltham, MA, USA).

## Histological assessment

Lung tissues were placed in 4% paraformaldehyde solution and subsequently paraffin embedded. Then, lung sections were subjected to hematoxylin-eosin (H&E) and Masson's trichrome staining for the assessment of histological changes. The presence of collagen deposition in lung sections was used to assess the degree of lung fibrosis as previously described (*Ashcroft, Simpson & Timbrell, 1988*; *Ding et al., 2019*).

## Determination of Iron

The iron content in lung tissues was measured by an IRON detection kit (MLBio, Inc., Shanghai, China). After adding samples and reagents to the 96-well plate, and measured the optical density value of samples at a wavelength of 562 nm with a microplate reader (Enspire, PerkinElmer, USA).

## Assessment of oxidative stress in lung tissues and BEAS-2B cells

The concentrations of 4-hydroxynonenal (4-HNE) were measured with a 4-HNE ELISA kit (MLBio, Inc., Shanghai, China) according to the protocol. Samples absorbance was measured at 450 nm with a microplate reader (Enspire; PerkinElmer) and calculated according to a standard curve. The GSH/GSSG ratio was determined by a commercial detection kit (Nanjing Jiancheng Bio-engineering, Inc., Nanjing, China) according to the manufacturer's protocol. The optical density value of sample was determined at a wavelength of 405 nm with a microplate reader (Enspire; PerkinElmer).

## Determination of nuclear translocation of Nrf2 in BEAS-2B cells

Immunofluorescence staining of Nrf2 was performed to determine nuclear translocation of Nrf2 in BEAS-2B cells. Cells were grown in 12-well plates and exposed to $PM_{2.5}$ and

Que for 24 hours. After washed twice with precooling PBS, cells were fixed with 4% paraformaldehyde at room temperature. Next, cells were permeabilized with 0.1% Triton X-100, and then incubated with Nrf2 (1:200, Proteintech Bio, Inc., Wuhan, China) primary antibody overnight at 4 °C. Subsequently, the cellular nucleus was stained with DAPI. The fluorescence images were visualized and photographed using the LSM 900 Meta laser scanning confocal microscope (Carl Zeiss, Oberkochen, Germany).

### Immunoblotting
We extracted total proteins of lung tissue and cells using RIPA lysis buffer protein extraction reagent (Beyotime Biological Co., Ltd., Shanghai, China). Protein samples underwent 10% SDS-PAGE gel electrophoresis under constant pressure of 220 V and then transferred to a polyvinylidene difluoride membrane under constant pressure of 110 V for 120 min. At room temperature, the membranes were blocked with 5% notfat milk for 2 h and then incubated with primary antibodies including anti-GPX4 (diluted 1:500), anti-ACSL4 (diluted 1:500), anti-GAPDH (diluted 1:1000), anti-Nrf2 (diluted 1:500), anti-Keap1 (diluted 1:500), anti-PCNA (diluted 1:500), anti-TGF-$\beta$1 (diluted 1:500) and anti-Collagen-I (diluted 1:500) at 4 °C overnight. Then, the blots were incubated with the goat anti-rabbit secondary antibody. The protein expression levels were visualized using a Tanon ECL detection system (Shanghai, China).

### Statistical analysis
The data are showed as the mean $\pm$ standard deviation (SD) and data analysis was carried out using SPSS 25.0. One-way ANOVA followed by post hoc analysis was used to compare two or more groups. $P < 0.05$ indicated a statistically significant difference.

## RESULTS

### Que attenuated PM$_{2.5}$-induced lung fibrosis and the inflammatory response in the lungs of mice
H&E staining of lung tissues showed obvious morphologic changes, including thickening of the alveolar septum and increasing of inflammatory cell in the pulmonary interstitium (Fig. 1A). Fig. 1B showed that lung collagen deposition was increased in the PM$_{2.5}$-exposed mice, but collagen deposition was suppressed by Que supplementation. Moreover, PM$_{2.5}$ significantly increased lung fibrosis scores in mice (Fig. 1C, mean $= 3.05$, one-way ANOVA, $F = 26.522$, $p = 0.000$, $n = 5$, $df = 19$). Moreover, lung fibrosis scores in the PM$_{2.5}$-exposed mice were reduced by Que supplementation (Fig. 1C, mean $= 3.05$, one-way ANOVA, $F = 26.522$, $p = 0.009$, $n = 5$, $df = 19$). No significant difference in lung weight/body weight was observed among the four groups (Fig. 1D, mean $= 4.00$, one-way ANOVA, $F = 1.159$, $p = 0.350$, $n = 6$, $df = 23$). Our results showed that the contents of IL-1$\beta$, IL-6 and TNF-$\alpha$ in the BALF of PM$_{2.5}$-exposed mice were obviously higher than those in the BALF of control mice (Figs. 1E–1G, mean $= 32.53/46.30/29.96$, one-way ANOVA, $F = 107.511/128.133/47.439$, $p = 0.000$, $n = 6$, $df = 23$). In addition, Que supplementation markedly decreased the contents of IL-1$\beta$, IL-6 and TNF-$\alpha$ in the BALF of mice challenged with PM$_{2.5}$. Taken together, our data showed that Que exerts antifibrotic and anti-inflammatory effects on the lungs of PM$_{2.5}$-challenged mice.

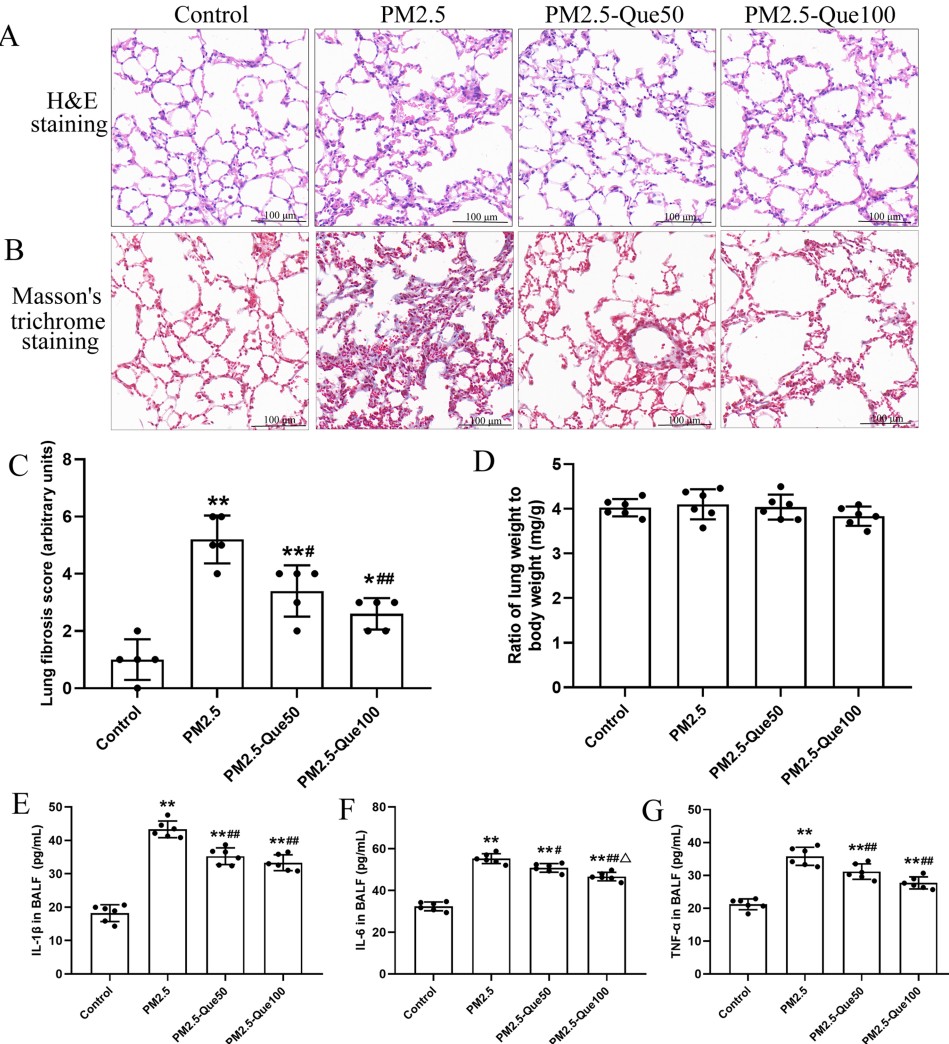

**Figure 1** **Que alleviated PM$_{2.5}$-induced lung fibrosis and inflammation in mice.** (A) H&E staining of lung tissues (scale bars =100 µm). (B) Masson's trichrome staining of lung tissues. (C) Lung fibrosis scores ($n = 6$). (D) Lung weight/body weight ($n = 6$). (E) IL-1$\beta$ in BALF ($n = 6$). (F) IL-6 in BALF ($n = 6$). (G) TNF-$\alpha$ in BALF ($n = 6$). Data are expressed as the mean ± SD. *, $P < 0.05$ and **, $P < 0.01$ compared with the control group. #, $P < 0.05$ and ##, $P < 0.01$ compared with the PM$_{2.5}$ group. △, $P < 0.05$ compared with the PM$_{2.5}$-Que/L group.

## Que reduced lipid peroxidation production and inhibited ferroptosis in lung tissues of PM$_{2.5}$-exposed mice

We measured the lipid peroxidation production 4-HNE and GSH/GSSG ratio in lung tissues. As presented in Figs. 2A–2B, PM$_{2.5}$ obviously increased 4-HNE content and decreased GSH/GSSG ratio in the lung tissues, and Que markedly reversed these above parameters in the lung tissues of PM$_{2.5}$-challenged mice (mean = 7.02/3.13, one-way ANOVA, $F = 39.451/100.593$, $p = 0.000$, $n = 6$, $df = 23$). To study the inhibitory effect of Que on ferroptosis in lung, we measured the protein expression of ferroptosis markers and the iron content in lung tissues. PM$_{2.5}$ exposure significantly increased iron content in lung

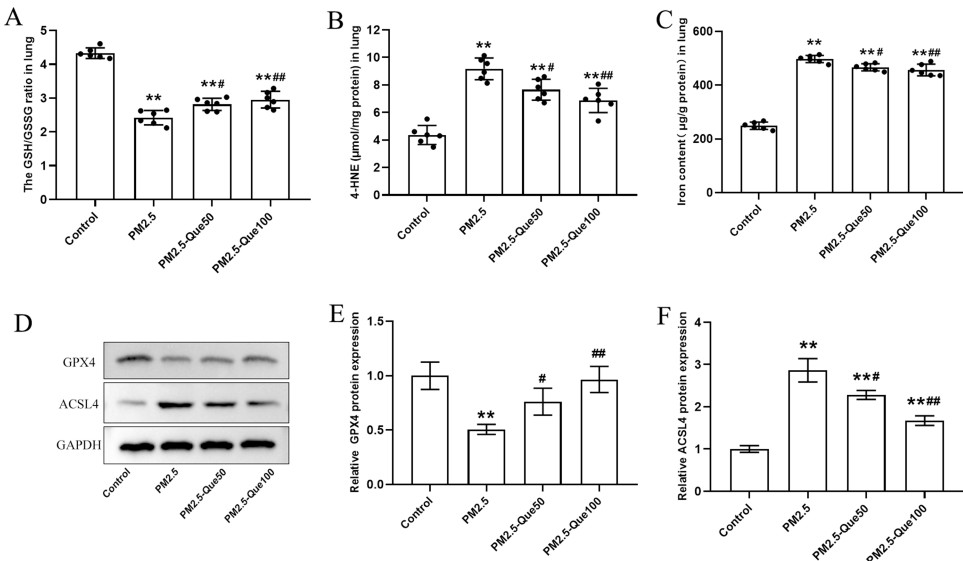

**Figure 2  Que reduced lipid peroxidation and iron content and inhibited ferroptosis in the lung tissues of PM$_{2.5}$-exposed mice.** (A–B) The levels of GSH/GSSG and 4-HNE in lung tissues ($n = 6$). (C) Iron content in lung tissues ($n = 6$). (D) Representative Western blot images of GPX4 and ACSL4 expression. (E–F) The expression of ferroptosis-related proteins (GPX4 and ACSL4) in lung tissues ($n = 3$). Data are expressed as the mean $\pm$ SD. *, $P < 0.05$ and **, $P < 0.01$ compared with the control group.#, $P < 0.05$ and ##, $P < 0.01$ compared with the PM$_{2.5}$ group.

tissues, which was reduced by Que treatment (Fig. 2C, mean = 4.17, one-way ANOVA, $F = 302.453$, $p = 0.000$, $n = 6$, $df = 23$). PM$_{2.5}$ exposure decreased GPX4 and increased ACSL4 in lung tissues, indicating that PM$_{2.5}$ has been activated ferroptosis in the lung tissues (Figs. 2D–2F, mean = 0.809/1.953, one-way ANOVA, $F = 19.317/76.463$, $p = 0.001/0.000$, $n = 3$, $df = 11$). Furthermore, Que treatment reversed the downregulated GPX4 and the upregulated ACSL4 in PM$_{2.5}$-challenged mice (Figs. 2D–2F). Our results demonstrated that Que treatment could suppress ferroptosis in the lung of PM$_{2.5}$-challenged mice, exerting strong anti-ferroptotic effects.

## Que activated Nrf2 in lung tissues of PM$_{2.5}$-exposed mice

To clarify the mechanism by which Que inhibits ferroptosis in lung tissues *in vivo*, we further determined the protein expression of nuclear Nrf2 and Keap1 in lung tissues. We found PM$_{2.5}$ apparently downregulated the nuclear Nrf2 protein expression and upregulated the Keap1 protein expression in lung tissues (Figs. 3A–3C, mean = 1.060/1.271, one-way ANOVA, $F = 47.922/27.542$, $p = 0.000/0.000$, $n = 3$, $df = 11$). In addition, Que treatment decreased the nuclear Nrf2 protein and decreased Keap1 protein expression in the lung of PM$_{2.5}$-treated mice (Figs. 3A–3C). Our data suggested that Que treatment apparently activated Nrf2 in the lung tissues of PM$_{2.5}$-treated mice.

## Que reduced the expression of TGF-$\beta$1 and Collagen-I in lung

To explore the beneficial function of Que on lung fibrosis, the protein expressions of TGF-$\beta$1 and Collagen-I were measured. PM$_{2.5}$ obviously increased the TGF-$\beta$1 protein

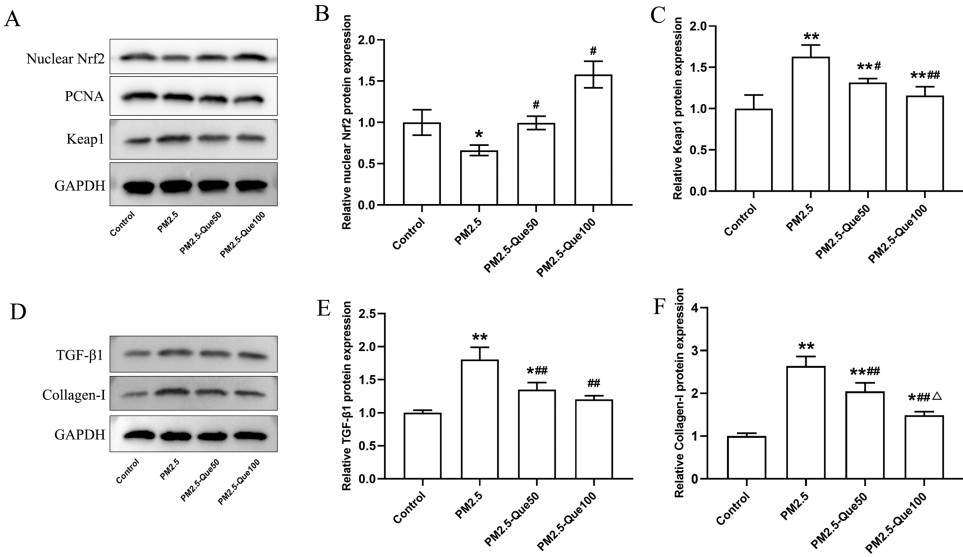

**Figure 3** **Que activated Nrf2 and reduced the expression of TGF-$\beta$1 and Collagen-I in lung tissues of PM$_{2.5}$-exposed mice.** (A) Representative Western blot images of Nrf2 and Keap1 expression. (B) The protein expression of Nrf2 in the nucleus in lung tissues ($n = 3$). (C) The protein expression of Keap1 in the cytoplasm in lung tissues ($n = 3$). (D) Representative Western blot images of TGF-$\beta$1 and Collagen-I expression. (E) The protein expression of TGF-$\beta$1 in lung tissues ($n = 3$). (F) The protein expression of TGF-$\beta$1 in lung tissues ($n = 3$). Data are expressed as the mean $\pm$ SD. Data are expressed as the mean $\pm$ SD. $\star$, $P < 0.05$ and $\star\star$, $P < 0.01$ compared with the control group. $^\#$, $P < 0.05$ and $^{\#\#}$, $P < 0.01$ compared with the PM$_{2.5}$ group. $^\triangle$, $P < 0.05$ compared with the PM$_{2.5}$-Que50 group.

expression and Collagen-I protein expression in the lung tissues and two doses of Que obviously reversed the increasing of TGF-$\beta$1 and Collagen-I caused by PM$_{2.5}$ (Figs. 3D–3E, mean $= 1.340/1.793$, one-way ANOVA, $F = 28.916/62.550$, $p = 0.000/0.000$, $n = 3$, $df = 11$). Taken together, Que may alleviate PM$_{2.5}$-induced lung fibrosis by reducing the expression of TGF-$\beta$1 and Collagen-I in the lung.

## Que reduced inflammatory response and TGF-$\beta$1 by inhibiting ferroptosis

BEAS-2B cells were exposed with Que (0, 5, 10, 20 and 40 μmol/L) and PM$_{2.5}$ (0–100 μg/mL) for 24 h. As shown in Fig. 4A, our results presented that PM$_{2.5}$ (50 μg/mL and 100 μg/mL) markedly decreased the viability of cells (mean $= 0.965$, one-way ANOVA, $F = 12.926$, $p = 0.000$, $n = 6$, $df = 29$). Thus, PM$_{2.5}$ (50 μg/mL) was used for *in vitro* study. The viability of BEAS-2B cells treated with Que (40 μmol/L) showed a significant decrease (Fig. 4B, mean $= 0.963$, one-way ANOVA, $F = 5.257$, $p = 0.003$, $n = 6$, $df = 29$), so Que at a dose of 20 μmol/L was used in the cell experiments.

To clarify the role of ferroptosis in PM$_{2.5}$-caused inflammation *in vitro,* we determined the ratio of GSH/GSSG and 4-HNE content in cell supernatants, the iron content, the expressions of ferroptosis markers and TGF-$\beta$1, and proinflammatory cytokines in cells exposed to PM$_{2.5}$. As presented in Figs. 4C–4E, PM$_{2.5}$ treatment significantly decreased the ratio of GSH/GSSG and increased the content of 4-HNE in cell culture

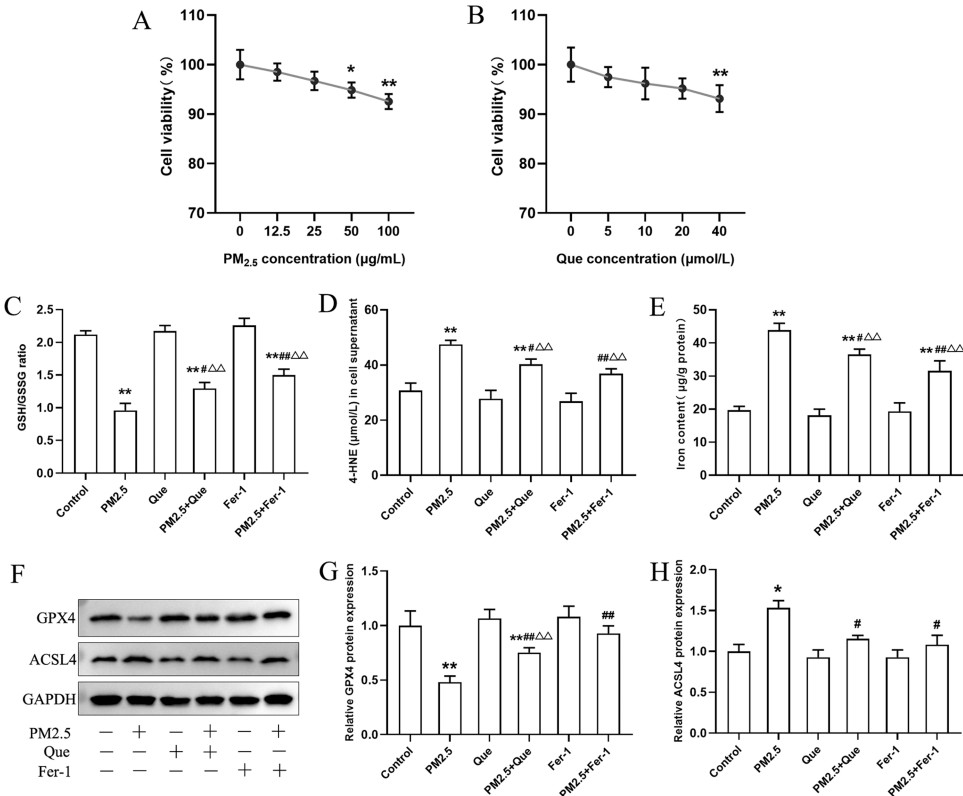

**Figure 4** **Que suppressed ferroptosis and ferroptosis-mediated inflammation in PM$_{2.5}$-treated BEAS-2B cells.** (A) The effect of PM$_{2.5}$ on BEAS-2B cell viability ($n = 3$). (B) Effect of Que on BEAS-2B cell viability ($n = 3$). (C–D) The ratio of GSH/GSSG and 4-HNE content in cell culture supernatants ($n = 3$). (E) Iron content in BEAS-2B cells ($n = 3$). (F) Representative Western blot images of ferroptosis-related protein (GPX4 and ACSL4) expression. (G–H) The protein expression of GPX4 and ACSL4 in BEAS-2B cells ($n = 3$). Data are expressed as the mean $\pm$ SD from three independent experiments. *, $P < 0.05$ and **, $P < 0.01$ compared with control cells. #, $P < 0.05$ and ##, $P < 0.01$ compared with the PM$_{2.5}$ group. $\triangle\triangle$, $P < 0.01$ compared with the corresponding control group.

supernatants and iron content in BEAS-2B cells, and these changes were reversed by Que treatment (mean = 1.718/35.032/28.219, one-way ANOVA, $F = 104.647/35.101/79.059$, $p = 0.000/0.000/0.000$, $n = 3$, $df = 17$). Moreover, PM$_{2.5}$ treatment decreased GPX4 expression and increased ACSL4 expression in BEAS-2B cells (Figs. 4F–4H, mean = 0.886/1.106, one-way ANOVA, $F = 37.774/24.820$, $p = 0.000/0.000$, $n = 3$, $df = 17$). In addition, both Que and Fer-1 significantly reversed the changes of the expressions of GPX4 and ACSL4 in BEAS-2B cells (Figs. 4F–4H). Importantly, PM$_{2.5}$ treatment obviously increased the contents of three proinflammatory cytokines in cell culture supernatants (Figs. 5A–5C, mean = 48.511/0.396/116.58, one-way ANOVA, $F = 113.701/73.407/124.116$, $p = 0.000/0.000/0.000$, $n = 3$, $df = 17$) and TGF-$\beta$1 protein expression in BEAS-2B cells (Figs. 5D–5E, mean = 1.206, one-way ANOVA, $F = 24.636$, $p = 0.000$, $n = 3$, $df = 17$). Furthermore, both Que and Fer-1 obviously reduced the contents of proinflammatory cytokines in cell culture supernatants and the TGF-$\beta$1 protein expression in PM$_{2.5}$-treated

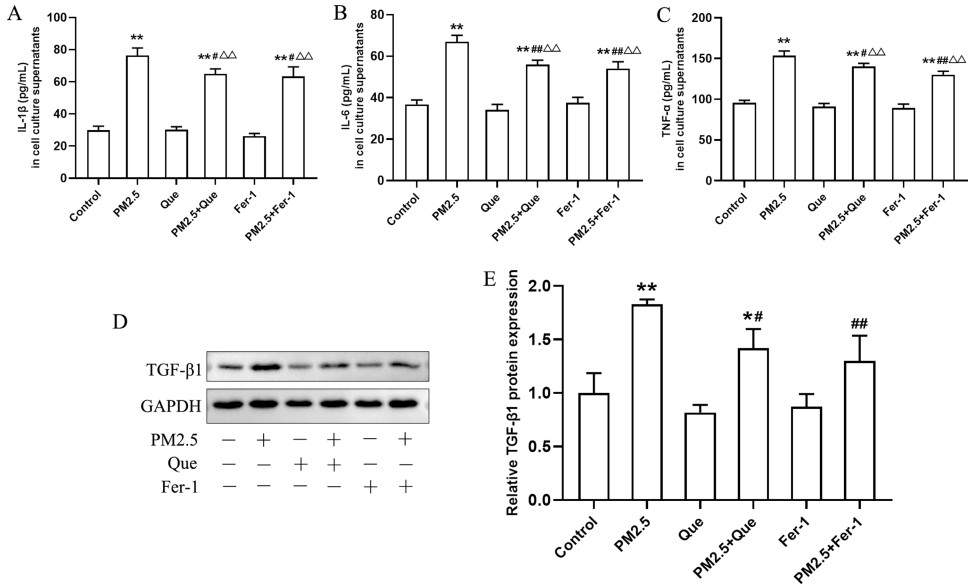

**Figure 5** **Que reduced inflammatory cytokines in cell culture supernatants and TGF-$\beta$1 in PM$_{2.5}$-treated BEAS-2B cells.** (A–C) The content of IL-1$\beta$, IL-6 and TNF-$\alpha$ in cell culture supernatants ($n = 3$). (D) Representative Western blot images of TGF-$\beta$1 expression. (E) The protein expression of TGF-$\beta$1 in BEAS-2B cells ($n = 3$). Data are expressed as the mean $\pm$ SD from three independent experiments. *, $P < 0.05$ and **, $P < 0.01$ compared with the control cells. #, $P < 0.05$ and ##, $P < 0.01$ compared with the PM$_{2.5}$ group. $\triangle\triangle$, $P < 0.01$ compared with their corresponding control group.

cells (Fig. 5). These results suggested that PM$_{2.5}$ increased the inflammatory response and TGF-$\beta$1 by triggering ferroptosis in BEAS-2B cells and that Que supplementation rescued these effects.

## Que inhibited ferroptosis by activating Nrf2 in BEAS-2B cells

We assessed the effect of Que and DMF on ferroptosis in BEAS-2B cells. As shown in Figs. 6A–6C, PM$_{2.5}$ treatment notably downregulated the protein expression of GPX4 and upregulated the protein expression of ACSL4 in BEAS-2B cells. Additionally, the decreased protein expression of GPX4 and the increased protein expression of ACSL4 in BEAS-2B cells were both rescued by Que treatment and DMF treatment (Figs. 6A–6C, mean = 0.961/1.538, one-way ANOVA, $F = 12.535/48.760$, $p = 0.000/0.000$, $n = 3$, $df = 17$). To confirm the mechanism by which Que inhibited PM$_{2.5}$-induced pulmonary ferroptosis, we further assessed whether Que could activate Nrf2 by DMF to induce ferroptosis in PM$_{2.5}$-treated BEAS-2B cells. As shown in Figs. 6D–6E, PM$_{2.5}$ significantly decreased nuclear Nrf2 in BEAS-2B cells, which was rescued by Que treatment (mean = 0.821, one-way ANOVA, $F = 45.430$, $p = 0.000$, $n = 3$, $df = 8$. Our results showed that the protein expression of nuclear Nrf2 and Keap1 was apparently decreased in PM$_{2.5}$-treated BEAS-2B cells compared with the control group (Figs. 6F–6H, mean = 1.422/2.190, one-way ANOVA, $F = 33.875/67.313$, $p = 0.000/0.000$, $n = 3$, $df = 17$). Moreover, the decreased protein expression of nuclear Nrf2 and the increased protein expression of Keap1 in PM$_{2.5}$-treated BEAS-2B cells were reversed by Que treatment and DMF treatment (Figs.

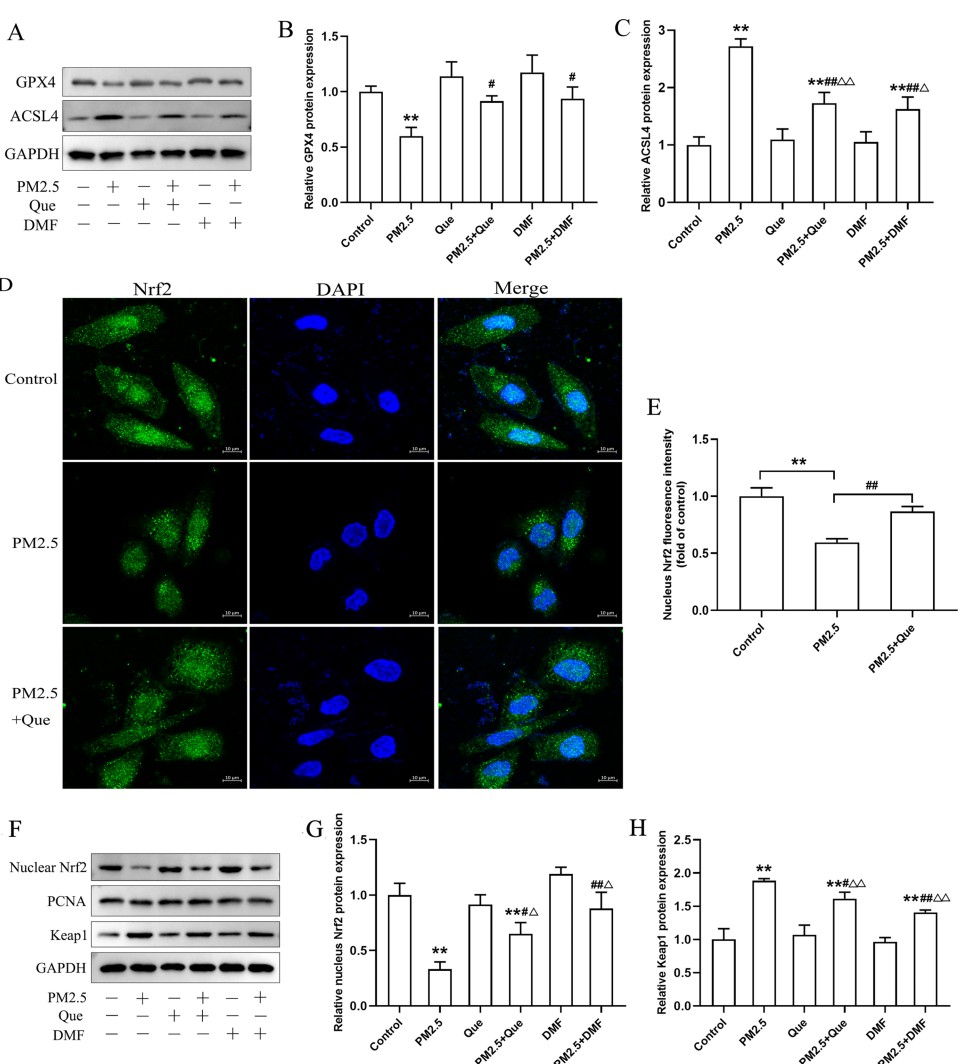

**Figure 6  Que suppressed ferroptosis in PM$_{2.5}$-treated BEAS-2B cells by activating Nrf2 *in vitro*.** (A) Representative Western blot images of GPX4 and ACSL4 expression. (B–C) The protein expression of ferroptosis-related proteins (GPX4 and ACSL4) in BEAS-2B cells ($n = 3$). (D) Fluorescence images of Nrf2 in BEAS-2B cells (scale bars =10 μm). (E) Fluorescence intensities of nuclear Nrf2 in cells ($n = 3$). (F) Representative Western blot images of nuclear Nrf2 and Keap1. (G) Protein level of nucleus Nrf2 ($n = 3$). (H) Protein expression of Keap1 in the cytoplasm ($n = 3$). Data are expressed as the mean ± SD from three independent experiments. **, $P < 0.01$ compared with control cells. #, $P < 0.05$ and ##, $P < 0.01$ compared with the PM$_{2.5}$ group. $^{\triangle}$, $P < 0.05$ and $^{\triangle\triangle}$, $P < 0.01$ compared with their corresponding control group.

6F–6H). Taken together, these data suggest that Que may inhibit ferroptosis by activating the Keap1-Nrf2 signaling pathway in PM$_{2.5}$-treated BEAS-2B cells.

# DISCUSSION

In this study, we found that Que supplementation alleviated PM$_{2.5}$-induced murine lung inflammation and fibrosis by decreasing ferroptosis. Mechanistically, we revealed that

Que treatment decreased inflammatory cytokines and the profibrotic factor TGF-$\beta$1 by activating Nrf2-Keap1 to suppress ferroptosis in lung epithelial cells. In summary, our study demonstrated the beneficial function of Que supplementation on PM$_{2.5}$-induced lung injury and the potential mechanism.

The toxicological effects of ambient PM$_{2.5}$ on the lung have been widely studied; however, the role of ferroptosis in PM$_{2.5}$-induced lung injury is still unclear. Ferroptosis is a newly identified type of cell death (*Jiang, Stockwell & Conrad, 2021b*; *Stockwell et al., 2017*). Recent studies have shown that the pathophysiological processes of many diseases are driven by ferroptosis (*Li et al., 2020b*). A previous study indicated that ferroptosis was increased in lung tissues of PM-exposed murine models and cell injury models (*Wang et al., 2022b*). Moreover, Que has been reported to effectively inhibit ferroptosis *via* different mechanisms to alleviate acute kidney injury and high-fat diet-caused hepatic lipotoxicity (*Jiang et al., 2022*; *Wang et al., 2021*). Here, our data demonstrated that PM2.5 markedly caused iron overload, triggered ferriptosis in lung and BEAS-2B cells, and these changes were reversed by Que. Here, we demonstrated that Que treatment could inhibit ferroptosis in the lung tissues of PM$_{2.5}$-exposed mice and PM$_{2.5}$-exposed BEAS-2B cells. Moreover, we observed that chronic exposure to PM$_{2.5}$ led to lung fibrosis. Therefore, we next investigated the underlying mechanism by which ferroptosis occurs in PM$_{2.5}$-induced lung injury and the protective mechanism of Que on lung fibrosis in PM$_{2.5}$-exposed mice.

TGF-$\beta$1 is one of the most notable and important profibrogenic factors that accelerates epithelial-mesenchymal transition, promotes fibroblast proliferation, increases extracellular matrix deposition, activates profibrotic pathways (*Woodcock et al., 2019*) and causes lung fibrosis (*Saito, Horie & Nagase, 2018*). Chronic PM$_{2.5}$ instillation could induce lung inflammation and pulmonary fibrosis by activating TGF-$\beta$1 in mice (*Xu et al., 2021*). In addition, ferroptosis inhibitors alleviate radiation-induced lung fibrosis by downregulating TGF-$\beta$1 in mice (*Li et al., 2019*). In v*ivo* and *in vitro* experiments showed that PM$_{2.5}$ increased the level of TGF-$\beta$1 and that inhibition of ferroptosis reduced the level of TGF-$\beta$1, suggesting that PM$_{2.5}$ could trigger ferroptosis to increase TGF-$\beta$1, which further causes lung fibrosis. Notably, Que alleviated PM$_{2.5}$-induced lung fibrosis *via* inhibition of ferroptosis in mice.

Ferroptosis plays important roles in the regulation of inflammation and oxidative stress in the pathogenesis of cardiovascular diseases (*Yu et al., 2021*). *Li et al. (2021)* reported that ferroptosis could mediate inflammation in lipopolysaccharide-treated BEAS-2B cells. A previous study reported that ferroptosis mediates inflammation in a lipopolysaccharide-induced acute respiratory distress syndrome murine model (*Wang et al., 2022a*). Due to the critical role of ferroptosis in mediating inflammation, the relationship between inflammation and ferroptosis in PM$_{2.5}$-induced lung injury was investigated. In an *in vivo* study, we observed that chronic and low concentrations of PM$_{2.5}$ exposure induced lung inflammation. *In vitro*, we found that inhibition of ferroptosis by treatment with Que and Fer-1 significantly decreased inflammatory in BEAS-2B cells treated with PM$_{2.5}$, indicating that ferroptosis plays a vital role in the regulation of PM$_{2.5}$-induced lung inflammation.

Nrf2-Keap1 signaling is recognized as an important endogenous antioxidative stress pathway that defends against oxidative and electrophilic stresses (*Yamamoto, Kensler &*

*Motohashi, 2018*). Nrf2 (a master regulator of the cellular antioxidant response) translocates to the nucleus and subsequently activates the transcription of antioxidant response genes when cells challenged with stress conditions (*Baird & Yamamoto, 2020*). Some previous studies proved that activation of the Keap1-Nrf2 pathway reduced ferroptosis in acute lung injury (*Li et al., 2021*; *Qiang et al., 2020*). Nrf2 could mediate the expression of genes involved in iron homeostasis and lipid peroxides in the ferroptotic process (*Dodson, Castro-Portuguez & Zhang, 2019*; *Kuang et al., 2020*). Considering that Nrf2 plays a critical role in mediating ferroptosis, we further investigated the role of Nrf2 in regulating lung injury-associated ferroptosis induced by $PM_{2.5}$ exposure and explored whether Que could activate Nrf2 to suppress ferroptosis in lung tissues. In addition, $PM_{2.5}$ could inhibit the activity of the Nrf2 signaling pathway in lung tissues in an allergic rhinitis mouse model (*Piao et al., 2021*). Our study found that $PM_{2.5}$ exposure reduced Nrf2 and triggered ferroptosis in the lung tissues of mice, which were rescued by Que treatment. Additionally, both the Nrf2-specific agonist and Que treatment significantly exerted anti-ferroptotic effects on $PM_{2.5}$-challenged BEAS-2B cells. Taken together, Que suppressed ferroptosis through causing Nrf2 activation in the lungs of $PM_{2.5}$-treated mice.

## CONCLUSIONS

In summary, our study first provides evidence that low concentrations of ambient $PM_{2.5}$ may induce lung inflammation and fibrosis through activating ferroptosis in mice. In addition, Que treatment protected against $PM_{2.5}$-caused lung injury by activating Nrf2-Keap1 to suppress ferroptosis in epithelial cells. These findings confirmed anti-ferroptosis by Que supplementation may be a novel therapeutic way to air pollution-caused lung injury.

**Abbreviations**

| | |
|---|---|
| **$PM_{2.5}$** | fine particulate matter |
| **WHO** | The World Health Organization |
| **Que** | Quercetin |
| **AVMA** | American Veterinary Medical Association |
| **BALF** | bronchoalveolar lavage fluids |
| **IL-1$\beta$** | interleukin-1$\beta$ |
| **IL-6** | interleukin-6 |
| **H&E** | hematoxylin-eosin |
| **4-HNE** | 4-hydroxynonenal |

## ACKNOWLEDGEMENTS

We wish to thank Ding Jinfeng for his help of data processing.

### Funding

This work was supported by the Development Project of Yancheng Medical Science and Technology Project (No. YK2021095) and the Scientific and Technological Innovation Team of Jiangsu Vocational College of Medicine (20234304). The funders had no role in study design, data collection and analysis, decision to publish, or preparation of the manuscript.

### Grant Disclosures

The following grant information was disclosed by the authors:
The Development Project of Yancheng Medical Science and Technology Project: YK2021095.
Scientific and Technological Innovation Team of Jiangsu Vocational College of Medicine: 20234304.

### Competing Interests

The authors declare there are no competing interests.

### Author Contributions

- Shibin Ding conceived and designed the experiments, performed the experiments, analyzed the data, prepared figures and/or tables, authored or reviewed drafts of the article, and approved the final draft.
- Jinjin Jiang conceived and designed the experiments, performed the experiments, prepared figures and/or tables, authored or reviewed drafts of the article, and approved the final draft.
- Yang Li analyzed the data, authored or reviewed drafts of the article, and approved the final draft.

### Animal Ethics

The following information was supplied relating to ethical approvals (*i.e.,* approving body and any reference numbers):

The Ethics Committee at the Jiangsu Vocational College of Medicine provided full approval for this research (ethics number: LLSQ-202104160001)

### Data Availability

The raw measurements are available in the Supplementary Files.

### Supplemental Information

Supplemental information for this article can be found online at http://dx.doi.org/10.7717/peerj.16703#supplemental-information.

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
