# Peer review of "Quercetin alleviates PM2.5-induced chronic lung injury in mice by targeting ferroptosis"

_PeerJ, doi:10.7717/peerj.16703_

## Round 0.1 · original submission · Major Revisions

Dear Dr. Ding,

Thank you for your submission to PeerJ. We have received an elaborate review report for your manuscript from a panel of expert reviewers and we invite you to resubmit the manuscript addressing all the comments raised during the review. If you choose to resubmit the manuscript after addressing the reviewer's comments, the following critical points must be addressed in order to reconsider your manuscript further-

1. Please quantify all H&E, Masson’s trichrome and immunofluorescence images and specify how many images from each group was quantified and perform statistical analysis.

2. Please elaborate the materials and methods with enough detail that anyone would be able to follow your procedures and/or repeat your experiment.

3. In methods, the authors write: “All mice were randomly (random number table method) assigned into four groups of 10 mice”- were there 10 mice/group? In the results data from 5-6 mice is shown. Please explain! If experiments were started with 10 mice /group, data from all 10 mice must be shown. Source files for figures can be provided if the figures do not show data for all animals in the group due to overlap.

Himangshu Sonowal
Academic Editor
PeerJ Life & Environment

**Language Note:** The review process has identified that the English language must be improved. PeerJ can provide language editing services - please contact us at copyediting@peerj.com for pricing (be sure to provide your manuscript number and title). Alternatively, you should make your own arrangements to improve the language quality and provide details in your response letter. – PeerJ Staff

·

Basic reporting

NA

Experimental design

Manuscript 88491-v0
In this study authors have investigated how Quercetin acts as a protective role against PM2.5-induced lung injury by activating Nrf2-Keap1 via suppressing ferroptosis in epithelial cells. Study design is good, but I have some issues about this article. Authors should go through the following comments.
1. They have used only 10 mice in this experiment for four groups. They should use a minimum of four animals in each group (Total 16 mice).
2. Authors should perform a time dependent assay (MTT assay) for determining the LD50 of PM2.5.
3. Authors should mention the pre- or pos- or simultaneous- treatment Quercetin with PM2.5.
4. Authors should include the bar diagram for PCNA expression (Figure 3A).

Validity of the findings

NA

Additional comments

NA

Reviewer 2 ·

Basic reporting

1. The whole manuscript needs excessive English editing.
2. Add the list of abbreviations.
3. Line 25………….route of quercetin.administeration
4. The abstract section is too long.
5. The experimental design is not straightforward regarding the dose references of QUE, PM2.5, route of administration, period………etc.
6. Line 105 After exposure, what is the period of exposure
7. a 1% Pentobarbital……..need a reference
8. PM2.5 collection, more details, and references
9. Lung bronchoalveolar lavage of the mice, more information and references
10. Why did the authors choose this type of cell, BEAS-2B cells?
11. Determination of pro-inflammatory cytokines………….need more details for the method used with steps
12. Assessment of oxidative stress in lung tissues and BEAS-2B cells………. need more information for the technique used with steps
13. Line 140……………… We extracted total proteins using RIPA lysis buffer…which protein of the cell or lung tissue
14. Immunoblotting needs to be rewritten again concerning the details, and the antibody dilution of each one used
15. Line 145…….. the blots were incubated with the specific secondary antibody…..details about it?
16. In general, the material and methods need to be rewritten again. It is so Simple, inaccurate, and missing a lot.
17. Data of food and water should be added.
18. The Collection of Blood and Organs subtitles are strange…. need to be mentioned.
19. The estimation of biochemical parameters should be written in detail regarding the methods and the spectrophotometer used.
20. The experimental design is not appropriate, and it should be rewritten again and a graphical abstract for the experimental design.
21. How the authors examined the data for homogeneity
22. Regarding the cell culture .., need more details about the method
23. Figure 4 ligands not consistent with the figure
24. The effect of PM2.5 on BEAS-2B cell viability.
25. Effect of Que on BEAS-2B cell viability
26. : What explains why cell viability decreased with increased QUE concentration?
27. Fluorescence images of Nrf2………….where the method used?
28. Determination of Iron………..details method?

Experimental design

In general, the material and methods need to be rewritten again. It is so Simple, inaccurate, and missing a lot.

Validity of the findings

no comment'

Additional comments

no comment'

Reviewer 3 ·

Basic reporting

no comment

Experimental design

no comment

Validity of the findings

no comment

Additional comments

In this manuscript, the authors summarized and discussed Nrf2 is involved in ferroptosis in PM2.5-induced lung injury, and quercetin can alleviate these adverse effects via activating Nrf2-Keap1 signaling pathway, on this basis, they further investigated the effect and potential mechanism of chronic exposure to low-concentrations of ambient PM2.5 on lung injury and the beneficial functions and possible molecular mechanisms of Que treatment on PM2.5 exposure-caused lung toxicology.This work provides new insight and opinion into the development of quercetin-mediated ferroptosis suppression provides a new therapeutic target for PM2.5-induced chronic lung injury. The manuscript is well-organized and clearly stated. There are some problems,which must be solved before it is considered for publication.
Minor
1.the role of Quercetin in the lung-related diseases is need to be added in the part of introduction
2.The figure 4 can add the fluorescence of the ferroptosis-related proteins (GPX4 and ACSL4) in BEAS-2B cells.

---

## Round 0.2 · Major Revisions

Dear Dr. Ding,
Thank you for submitting the revised manuscript. Reviewer 2 has raised a few more concerns and I request you to to kindly provide responses to the comments raised by reviewer 2 and re-submit the manuscript for review.

Best regards,
Himangshu Sonowal

·

Basic reporting

N/A

Experimental design

Authors have properly addressed my comments properly. Now this article can go for publication.

Validity of the findings

N/A

Additional comments

N/A

Reviewer 2 ·

Basic reporting

The authors have improved the manuscript. However, some points need to be considered.
1. The doses of Que should be referenced.
2. The description of the mice used should be added.
3. The experimental period depends on what?
4. PM2.5 concentration (35 μg/m3), suggested by the WHO air quality guidelines. Need references.
5. The sampling section should be added.
6. The cell viability result is incompatible with other results concerned with increased QUE concentration; please explain.
7. The data homogeneity needs more explanation…what test is used for it, and mention how the authors make the sample size?
8. These experiments need further investigation on the gene expression and molecular levels

Experimental design

Original primary research within Aims and Scope of the journal.

Validity of the findings

Good

Reviewer 3 ·

Basic reporting

no comment

Experimental design

no comment

Validity of the findings

no comment

Additional comments

no comment

---

## Round 0.3 · Minor Revisions

Dear Dr. Ding,

The following comments are raised by the reviewer and I request you to address these concerns raised by the reviewer before the manuscript can be considered further.

The comments can be addressed by including a detailed description of materials and methods.

The following are the points raised by the reviewer:

1. The authors should add the references for the doses of Que not according to their pre-experiment
2. What is the basis for choosing the period of exposure
3. How the authors calculate the sample size
4. Add a section on sampling
5. The answer of the authors to the data homogeneity needs more explanation
6. The doses of Que in cell viability need references

I hope that you will kindly address these at the earliest and resubmit the manuscript.

Thank you!

Best,
Himangshu Sonowal
Academic Editor

Reviewer 2 ·

Basic reporting

The authors respond to the majority of the comments but I have some concern about the following:
1. The authors should add the refernces for the doses of Que not according to their pre-experiment
2. What is the base for choose the period of exposure
3. How the authors calculate the sample size
4. Add section of sampling
5. The answer of the authors to the data homogeneity need more explain
6. The doses of Que in cell viability need references

Experimental design

Methods described with sufficient detail & information to replicate

Validity of the findings

All underlying data have been provided; they are robust, statistically sound, & controlled.

---

## Round 0.4 · accepted · Accept

Dear Dr. Ding,

Thank you for your submission to PeerJ and satisfactorily addressing the comments raised by the reviewers during review process. I am happy to inform you know that the manuscript is now recommended to be considered for publication.

Best regards,
Himangshu Sonowal
Academic Editor
PeerJ Life & Environment